# The Impact of COVID-19 on the Diagnosis and Treatment of Lung Cancer over a 2-Year Period at a Canadian Academic Center

Goulnar Kasymjanova [1,*] , Angelo Rizzolo [1], Carmela Pepe [1], Jennifer E. Friedmann [1], David Small [1], Jonathan Spicer [2] , Magali Lecavalier-Barsoum [1], Khalil Sultanem [1], Hangjun Wang [1], Alan Spatz [1], Victor Cohen [1] and Jason S. Agulnik [1]

[1] Peter Brojde Lung Cancer Centre, Jewish General Hospital, Montreal, QC H3T 1E2, Canada
[2] Division of Thoracic Thoracic Oncology, McGill University Health Centre, MUHC, Montreal, QC H4A 3S5, Canada
* Correspondence: gkasymja@jgh.mcgill.ca; Tel.: +1-514-340-8222

**Abstract:** Background: We have recently reported a 35% drop in new lung cancer diagnoses and a 64% drop in lung cancer surgeries during the first year of the pandemic. Methods: The target population was divided into three cohorts: pre-COVID-19 (2019), first year of COVID-19 (2020), and second year of COVID-19 (2021). Results: The number of new lung cancer diagnoses during the second year of the pandemic increased by 75%, with more than 50% being in the advanced/metastatic stage. There was a significant increase in cases with multiple extrathoracic sites of metastases during the pandemic. During the first year of the pandemic, significantly more patients were treated with radiosurgery compared to the pre-COVID-19 year. During the second year, the number of radiosurgery and surgical cases returned to pre-COVID-19 levels. No significant changes were observed in systemic chemotherapy and targeted therapy. No statistical difference was identified in the mean wait time for diagnosis and treatment during the three years of observation. However, the wait time for surgery was prolonged compared to the pre-COVID-19 cohort. Conclusions: The significant drop in new diagnoses of lung cancer during the first year of the pandemic was followed by an almost two-fold increase in the second year, with the increased rate of metastatic disease with multiple extra-thoracic site metastases. Limited access to surgery resulted in the more frequent use of radiosurgery.

**Keywords:** lung cancer; COVID-19; wait times; pattern of treatment



## 1. Introduction

The COVID-19 burden unintentionally posed a major risk to cancer care globally, causing disruptions or delays in services as a result of stressed health systems. This directly affected screening, diagnosis, and treatment of cancer patients. An expert panel of 24 members, including pulmonologists, thoracic radiologists, and thoracic surgeons, have reviewed pre-COVID-19 guideline recommendations for lung screening and lung nodule evaluation [1]. The consensus was that it is appropriate to defer lung cancer screening and modify the evaluation of lung nodules due to risk from potential exposure and the need to relocate resources. Despite the revised recommendations, medical organisations experienced many failures during the COVID-19-era. More than 2500 studies from across the world were reporting hardship with cancer care delivery: due to increased risk of contracting COVID-19 and of mortality in cancer patients, especially in patients undergoing chemotherapy many cancer diagnostic tests and treatments were canceled or delayed because of COVID-19 [2–4]. We have recently reported a 35% drop in new lung cancer diagnoses and a 64% drop in lung cancer surgeries during the first year of the pandemic. Similarly, Hanna et al. reported a 34% and a 60% decrease in diagnosis and surgeries across all cancer in the province of Ontario [5]. The impact of COVID-19 on lung cancer treatment

outcomes is yet to come, but it is known that even a short delay in cancer diagnosis and treatment increases the risk of death by 10% [5–7]. In this study, we further investigate the impact of pandemics on the cancer diagnosis and care trajectory, extending the observation to the end of second year of COVID-19.

## 2. Materials and Methods

### 2.1. Study Population

This is a retrospective cohort study of patients diagnosed with lung cancer between March 2019 and February 2022 at the Peter Brojde Lung Cancer Center. The target population was divided into three cohorts:

- Cohort #1: Pre-COVID-19 (1 March 2019–29 February 2020);
- Cohort #2: 1st year of COVID-19 (1 March 2020–28 February 2021);
- Cohort #3: 2nd year of COVID-19 (1 March 2021–28 February 2022).

The study was approved by the Research Ethics Board (REB). Patients were identified from the electronic health record system and included in the analysis if they had a confirmed pathological diagnosis of lung cancer, known treatment characteristics (such as dates and type of treatment), and were followed at the Jewish General Hospital. Second opinions were excluded from the study population.

### 2.2. The Primary Objective

To investigate the impact of the COVID-19 pandemic on new lung cancer diagnoses.

### 2.3. Secondary Objectives

- To characterize treatment pattern changes in lung cancer during the COVID-19 pandemic;
- To evaluate the wait times before and during the COVID-19 era.

### 2.4. Data Collection

For this study, the following information was collected from electronic medical records:

- Demographics: age, sex, smoking history;
- Disease characteristics: stage, histopathological diagnosis, molecular testing results;
- Treatment history: referral date, type of first definitive treatment (FDT), start and end date of treatment;
- Diagnosis timing: referral date, date of first lung specialist consult, date of diagnosis;
- Survival data (to be reported later).

### 2.5. Definitions of Wait Times

The intervals investigated for this study are shown in Table 1, with the recommended wait times from existing guidelines [8–11].

### 2.6. Statistical Analysis

In this study, the mean, median, and ranges were used to summarize patient characteristics and wait time intervals. Binary wait time variables were used to calculate the proportion of patients who met the recommended wait times. Chi-square statistics were used to define the significance of the differences. A *p*-value of less than 0.05 is considered a significant difference. The dataset was locked on 28 February 2022. All statistical analysis was performed using the SPSS software.

**Table 1.** Recommended wait times for lung cancer patients.

| Wait Time | Mean Time (Days) | Guidelines |
|:---:|:---:|:---:|
| Referral → Lung cancer specialist | 14 | National Health Service [10] |
| Referral → Diagnosis | 30 | British Thoracic Society [8] |
| Referral → FDT [1] | 62 | National Health Service |

**Table 1.** *Cont.*

| Wait Time | Mean Time (Days) | Guidelines |
|---|---|---|
| Diagnosis → FTD | 30 | British Thoracic Society |
| DTT [2] → FTD | 31 | British Thoracic Society |
| Diagnosis → First chemotherapy | 28 | British Thoracic Society |
| Surgery consult → Surgery | 28 | British Thoracic Society |
| Radiation consult → First radiation therapy | 42 | RAND Corporation [9] |

[1] FTD—first definitive treatment, which is defined as the start of the treatment aimed at removing or eradicating cancer completely or at reducing tumor bulk (surgery, radiotherapy, or chemotherapy). [2] DTT—decision to treat, defined as the date the patient agrees with a treatment plan.

## 3. Results

### 3.1. Study Population

During the study period, a total of 500 patients were diagnosed with lung cancer. This included both non-small cell lung cancer (468 cases) and small cell lung cancer (32 cases). A total of 170 patients were included in 2019, 111 patients in 2020, and 219 patients in 2021 (Figure 1, Table 2). The overall number of diagnosed lung cancer cases decreased by 34.7% in the first year of the pandemic (Cohort 2). During the second year of the pandemic (Cohort 3), the number of new diagnoses increased by 97% compared to 2020 (cohort 2) and by 33% compared to 2019 (Cohort 1). After excluding all cases of second opinions, there were 130, 103, and 184 patients, respectively, who were treated at our center during the 3 years. This 417 patient were included in the study (Figure 1).

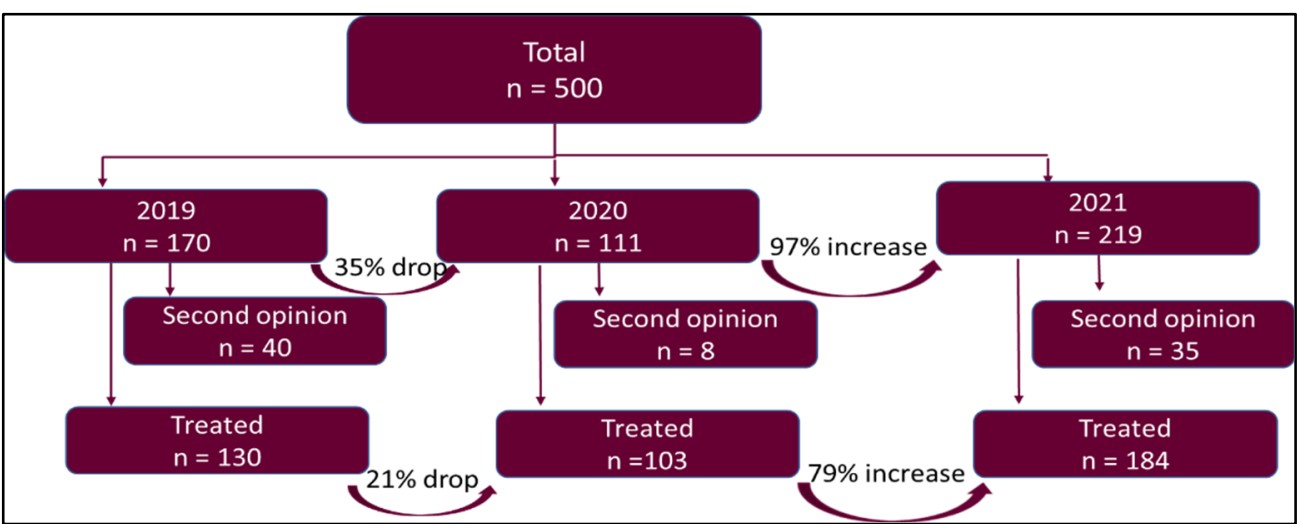

**Figure 1.** Patient flow chart.

Patients' characteristics are presented in Table 2. There was no statistical difference between the study cohorts. The mean age of the patients was similar. The majority were smokers with an advanced stage of lung cancer.

### 3.2. New Diagnosis of Lung Cancer

The 21% drop in new lung cancer diagnoses (after excluding second opinions) during the first year of the pandemic (Cohort 2) was followed by a 79% increase in the second year of the pandemic (Cohort 3) (Table 3). Although the proportion of advanced/metastatic stage of lung cancer remained the same for the three cohorts (Table 2), there was a significant increase in the proportion of M1c (multiple extrathoracic sites) cases in Cohorts 2 and 3 (57% and 51%) compared to Cohort 1 (31%), which suggested more advanced disease in the patients presenting during the pandemic (Table 4).

**Table 2.** Patients' characteristics.

| Characteristics | | 2019 $n = 130$ | 2020 $n = 103$ | 2021 $n = 184$ |
|---|---|---|---|---|
| Age (mean; range) | | 70 (40–96) | 71 (42–92) | 71 (41–92) |
| Sex (male/female) | | 73/57 | 56/47 | 88/96 |
| Cancer stage ($n$/%) | Early stage ($T_{1–3}N_{0–1}M0$) Early stage ($T_{1–3}N_{0–1}M0$) Locoregional ($T_{1–4}N_{2–3}M0$) Advanced/metastatic stage ($T_{any}N_{any}M_1$) | 42 (32) | 33 (35) | 55 (30) |
| | Locoregional ($T_{1–4}N_{2–3}M0$) | 20 (15) | 11 (10) | 29 (15) |
| | Advanced/metastatic stage ($T_{any}N_{any}M_1$) | 68 (53) | 59 (55) | 100 (55) |
| Smoking history ($n$/%) | Former/current smoker Non-smoker | 99 (76) | 74 (74) | 156 (85) |
| | Non-smoker | 31 (24) | 29 (26) | 28 (15) |
| Treatment type: ($n$/%) | FDT [1] | 110 (85)) | 84 (82) | 151 (84) |
| | PT [2] | 20 (15) | 19 (18) | 33 (16) |

[1] = first definitive treatment, [2] = palliative treatment.

**Table 3.** Lung cancer diagnoses in the three cohorts.

| Variables | 2019 | 2020 | 2021 |
|---|---|---|---|
| Number of new diagnoses treated | 130 | 103 | 184 |
| Average per month | 11 | 9 | 15 |
| Change in volume of new diagnoses compared to 2019 [a] | | −21% | +41% |
| Change in volume of new diagnoses compared to 2020 [b] | | | +79% |

[a]—2019 is the comparator, [b]—2020 is the comparator.

**Table 4.** Number of distant metastases.

| M Status | | 2019 $n = 68$ | 2020 $n = 59$ | 2021 $n = 100$ |
|---|---|---|---|---|
| M1a $n$ (%) | Pericardial/pleural effusion or contralateral lung | 20 (29) | 14 (24) | 31 (31) |
| M1b $n$ (%) | Single extrathoracic site | 27 (40) | 11 (19) | 18 (18) |
| M1c $n$ (%) | Multiple extrathoracic sites | 21 (31) | 34 (57) | 51 (51) |
| The two-tailed $p$ value | 2020 vs. 2019 | | 0.002 | |
| The two-tailed $p$ value | 2021 vs. 2019 | | | 0.013 |

*3.3. Treatment Pattern*

The type of first treatment received is presented in Table 5. Overall, 345 patients received their first definitive treatment (FDT): surgery, radiosurgery, or systemic treatment. Eighty-five percent of patients received FDT in 2019, and 82% and 84% in the first and second years of the pandemic, respectively (Table 2). During the first year of the pandemic (Cohort 2), significantly more patients were treated with radiosurgery as the first definitive treatment compared to the pre-COVID-19 year (Cohort 1). During the second year of the pandemic (Cohort 3), the number of radiosurgery cases returned to pre-COVID-19 levels ($p < 0.05$). In contrast, surgical cases dropped significantly in 2020 and returned to pre-COVID-19 levels in 2021. No significant changes were observed in systemic chemotherapy and targeted therapy ($p > 0.05$).

*3.4. Wait Times*

Tables 6 and 7 outline the mean wait times for each interval, and the proportion of patients meeting the recommended wait times. Despite the pandemic and the use of telemedicine, there were no significant delays in the first appointment with a lung cancer specialist, and 70% of patients were seen within the recommended 14-day target even

during the pandemic. (Table 6)There was no significant difference in wait time means for lung cancer diagnosis before and during the pandemic, as well as for first treatment ($p = 0.94$). Among patients who received FDT, 48%, 61% and 61% achieved the target wait time of 28 days in three cohorts, respectively (Table 7).

**Table 5.** Type of first definitive treatment (FTD).

| Type of FTD | | 2019 $n$ = 110 | 2020 $n$ = 84 | 2021 $n$ = 151 | *p*-Value |
|---|---|---|---|---|---|
| Surgery ($n$/%) | | 42 (38) | 24 (25) | 54 (36) | 0.01 |
| Radiosurgery ($n$/%) | | 8 (7) | 20 (21) | 12 (8) | 0.009 |
| | Total | 60 (54) | 40 (47) | 85 (56) | |
| Chemotherapy ($n$/%) | Standard systemic chemotherapy | 21 (35) | 12 (30) | 25 (29) | >0.05 |
| | Immunotherapy $\pm$ chemotherapy | 23 (37) | 16 (40) | 37 (44) | |
| | Targeted therapy | 17 (28) | 12 (30) | 23 (27) | |

**Table 6.** Mean wait times before and during two years of the COVID-19 pandemic.

| Interval (Days) | 2019 $n$ = 130 | | 2020 $n$ =103 | | 2021 $n$ = 184 | | *p*-Value |
|---|---|---|---|---|---|---|---|
| | $n$ | Mean (SD [1]) | $n$ | Mean (SD [1]) | $n$ | Mean (SD [1]) | |
| Referral $\rightarrow$ Lung cancer specialist (14) | 130 | 12 (14) | 103 | 11 (13) | 184 | 10 (11) | 0.67 |
| Referral $\rightarrow$ Diagnosis (30) | 130 | 59 (51) | 103 | 59 (67) | 184 | 65 (94) | 0.94 |
| Wait for the path report (14) | 130 | 8 (7.3) | 103 | 10 (12) | 184 | 8 (9) | 0.98 |
| Decision-to-treat to FDT [2] (31) | 130 | 52 (48) | 103 | 51 (61) | 151 | 43 (51) | 0.94 |
| Diagnosis to chemotherapy (28) | 60 | 38 (25) | 40 | 34 (24) | 85 | 43 (33) | 0.95 |
| Diagnosis to RT [3] (42) | 24 | 35 (30) | 38 | 46 (33) | 49 | 40 (53) | 0.31 |
| Surgical consult to surgery (28) | 42 | 64 (43) | 24 | 76 (83) | 54 | 77 (62) | 0.04 |
| Wait for molecular test results (7) | 142 | 22 (10) | 102 | 16 (9) | 218 | 26 (12) | 0.90 |

[1] Standard deviation, [2] first definitive treatments, [3] radiation treatments.

**Table 7.** Meeting the wait time standards.

| Interval (Days) | 2019 | 2020 | 2021 | *p*-Value |
|---|---|---|---|---|
| | Proportion (%) | | | |
| Referral $\rightarrow$ Lung cancer specialist (14) | 90/130 (69) | 74/103 (72) | 123/184 (67) | 0.35 |
| Referral $\rightarrow$ Diagnosis (30) | 52/130 (40) | 50/103 (48) | 82/184 (45) | 0.15 |
| Referral $\rightarrow$ FDT (62) | 55/110 (51) | 49/84 (58) | 91/151 (60) | 0.29 |
| Diagnosis $\rightarrow$ First systemic therapy [1] (28) | 24/60 (37) | 17/40 (42) | 34/85 (40) | 0.45 |
| Diagnosis $\rightarrow$ First targeted therapy (28) | 11/17 (65) | 5/12 (42) | 13/23 (57) | 0.43 |
| Surgical consult $\rightarrow$ Surgery (28) | 6/42 (14) | 8/24 (30) | 12/54 (22) | 0.12 |
| RadOnc [2] consult $\rightarrow$ Radiation treatment (42) | 17/24 (71) | 30/38 (79) | 35/48 (73) | 0.52 |

[1] Systemic treatment includes chemotherapy, IO, or a combination of the two, [2] radiation oncologist.

During COVID-19, wait time for surgery was prolonged compared to the pre-COVID-19 cohort ($p = 0.04$). The majority of patients failed to meet the 28 days target wait time from the first appointment with the surgeon. No statistical difference was observed in the type of surgery. VATS lobectomy was the most common: 27/46 (58%) in 2019, 16/26 (62%) in 2020, and 32/54 (59%) in 2021.

None of the other FDTs had any statistically significant delays during the COVID-19 pandemic. The mean wait time for definitive radiation was within the recommended 42 days' target time before and during the pandemic: 35 vs. 46 vs. 40 days, respectively (Table 6).

*3.5. Molecular Testing*

The total number of molecular tests dropped by 28% in 2020 (*n* = 102), but increased by 53% in 2021 (*n*= 218) compared to 2019 (*n* = 142). However, the turnaround time significantly increased for next-generation sequencing (NGS) and circulating tumor DNA (ctDNA) tests during the 2 years of the pandemic (Table 8).

**Table 8.** Mean wait time for molecular testing.

| Test | 2019 *n* = 130 | | 2020 *n* = 103 | | 2021 *n* = 184 | | *p*-Value |
|---|---|---|---|---|---|---|---|
| | *n* | Mean (SD [1]) Days | *n* | Mean (SD [1]) Days | *n* | Mean (SD [1]) Days | |
| NGS [2] | 66 | 15 (6) | 62 | 18 (9) | 117 | 31 (10 | 0.03 |
| ctDNA | 58 | 6 (10) | 32 | 10 (18) | 179 | 16 (17) | 0.05 |
| NanoString | 18 | 25 (11) | 27 | 27 (18) | 22 | 30 (12) | 0.50 |

[1] Standard deviation, [2] next-generation sequencing.

**4. Discussion**

The findings of our study confirm that pandemic-related delay and interruption of lung cancer care caused a 35% drop in new diagnoses of lung cancer during the first year of the pandemic. The reasons for such a drop were either patient-related (fear of contracting COVID-19, quarantining, or staying at home) or system-related (health care re-organization, closed primary care clinics, and limited guidelines to help with lung cancer care). Studies from Italy reported a 20–27% drop in lung cancer diagnoses as an impact of the COVID-19 lockdown [12,13].

As a result of the initial drop, the number of new cases increased by 79% in the second year, with more than 50% of them being in the advanced/metastatic stage. We observed the trend of a decline in new lung cancer diagnoses among males, and an increase in females. Similarly, Mangole et al. reported strong gender difference in the rate of lung cancer diagnosis during the first year of COVID-19 when compared to pre-COVID-19 year: a 17.6% drop in males and 14.1% increase in females [14]. Among the advanced/metastatic stage patients, there was a significant increase in the number of cases with multiple extrathoracic sites of metastases (M1c) during the first and second years of the pandemic. A similar large decrease in new lung cancer cases has been observed across Canada and internationally [5,7,15–22].

Despite the two-year pandemic and the strain put on hospital and medical staff, the majority of our patients were seen by the cancer specialist within the recommended wait time. No statistical difference was identified in the mean wait time for diagnosis and treatment during the three years of observation. Target wait time for FDT was achieved in the majority of our patients independently of the pandemic. However, more than 40% of patients failed to meet the target for the diagnosis-to-treatment interval. We recognize that wait time guidelines were created in the non-pandemic era and during the pandemic, with limited recourses, and these guidelines should be updated [11]. Olsson et al. reported before COVID-19 that median times to diagnosis (range 8–60 days) and times to treatment (range 30–84 days) often exceeded published recommendations [23]. A significant decrease and the longest delay in our study were observed for surgery. It is not surprising, as it has been reported that in the first COVID-19 waves, at least 21 million elective operations were canceled globally, partly due to concerns over postoperative COVID-19 infection and partly due to capacity issues within hospitals [24]. Riera et al. reported that 62 studies included in his systematic review identified 38 different categories of delay and disruption of services, including the number of cancer surgeries [6]. This might be explained by the COVID-19-related guidelines of each country. The ministry of health of Quebec in March 2020 implemented the protocol of systematically prioritizing resources away from surgical care to patients with COVID-19, which reduced surgical activities to a minimum, leaving operating rooms to run semi-urgent and urgent surgeries only [25]. Cancer treatment

delay during a pandemic is a problem for the health system worldwide, and the long-term consequences are yet to be determined.

As a result of the prioritization of available treatments, we observed a significant increase in radiosurgery for early-stage lung cancer during the first year of the pandemic. During the second year of COVID-19, the proportion of patients treated surgically and radio-surgically returned to the pre-pandemic level.

The large increase in new cases of lung cancer resulted in a significant increase in the amount of molecular testing in the second year, which significantly prolonged the wait time for molecular results and subsequently caused more delays in treatment.

## 5. Limitations

This study is, to our knowledge, the largest single-institution report comparing lung cancer trajectory before and during the COVID-19 pandemic in the province of Quebec. As with any retrospective chart review, we cannot determine the causation of diagnosis and treatment delay but only associations. The study group is not representative of the general population and is prone to selection bias. The results may not be generalizable to other institutions.

## 6. Conclusions

COVID-19 seems to have had a significant impact on our lung cancer patients for diagnoses and treatments. The significant drop in new diagnoses of lung cancer during the first year of the pandemic was followed by an almost two-fold increase in the second year, with an increased rate in metastatic disease with multiple extra-thoracic site metastases. Limited access to surgery resulted in the more frequent use of radiosurgery. Although the mean wait times for diagnosis and treatment were no different before and during the pandemic, the recommended targets were not achieved for half of the patients. This advocates for a revision of the existing recommendations. This study is still ongoing, and further data will be collected and analyzed to better understand the long-term impact of the COVID-19 pandemic on morbidity and mortality in the lung cancer patient population.

**Author Contributions:** Conceptualization, J.S.A. and G.K.; methodology, J.S.A.; software, G.K.; validation, G.K. and J.S.A.; formal analysis, G.K.; investigation, G.K. and J.S.A.; resources, J.S.A.; data curation, G.K.; writing—original draft preparation, G.K. and J.S.A; writing—review and editing, J.S.A., A.R., C.P., J.E.F., D.S., J.S., M.L.-B., K.S., H.W., A.S. and V.C.; visualization, J.S.A.; supervision, J.S.A.; project administration, G.K. and J.S.A.; funding acquisition, J.S.A. All authors have read and agreed to the published version of the manuscript.

**Funding:** This research received no external funding.

**Institutional Review Board Statement:** Protocol # is 2021-2655: "Lung cancer treatment pattern during COVID-19-era, wait times and outcomes. A retrospective chart review". The Final Authorization to conduct research at the Jewish General Hospital was granted by CIUSSS West-Central Montreal Research Ethics Board on 22 January 2021. The annual renewal was approved on 28 January 2022.

**Informed Consent Statement:** Patient consent was waived due to REASON of retrospective chart review type of study.

**Data Availability Statement:** Data available on request due to ethical restrictions.

**Conflicts of Interest:** The authors declare no conflict of interest.

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
