# Peer review of "The Impact of COVID-19 on the Diagnosis and Treatment of Lung Cancer over a 2-Year Period at a Canadian Academic Center"

_curroncol, doi:10.3390/curroncol29110684_

Round 1

Reviewer 1 Report

This is a very important research study that should be published. Aside from a rare grammatical or typographical error , the manuscript is very well written and requires no revisions.

The only critique that I can offer to the authors, for their possible revision, is the omission of a reference to a guideline article from the American College of Chest Physicians (ACCP) that in 2020, made very specific recommendations.

Chest. 2020 Jul; 158(1): 406–415.

Published online 2020 Apr 23. doi: 10.1016/j.chest.2020.04.020

PMCID: PMC7177089

PMID: 32335067

Management of Lung Nodules and Lung Cancer Screening During the COVID-19 Pandemic

CHEST Expert Panel Report

Peter J. Mazzone, MD, MPH, FCCP,a, Michael K. Gould, MD, FCCP,c Douglas A. Arenberg, MD, FCCP,d Alexander C. Chen, MD,f Humberto K. Choi, MD, FCCP,b Frank C. Detterbeck, MD, FCCP,g Farhood Farjah, MD, MPH,h Kwun M. Fong, MD,i Jonathan M. Iaccarino, MD,j Samuel M. Janes, PhD,k Jeffrey P. Kanne, MD, FCCP,l Ella A. Kazerooni, MD,e Heber MacMahon, MB, BCh,m David P. Naidich, MD, FCCP,n Charles A. Powell, MD, FCCP,o Suhail Raoof, MD, Master FCCP,p M. Patricia Rivera, MD, FCCP, ATSF,q Nichole T. Tanner, MD, MSCR, FCCP,r Lynn K. Tanoue, MD, FCCP,s Alain Tremblay, MDCM,t Anil Vachani, MD, FCCP,u Charles S. White, MD,v Renda Soylemez Wiener, MD, MPH,w,x and Gerard A. Silvestri, MD, FCCPy

The ACCP group specifically recommended

1.      that screening for lung cancer using CT scans should not be performed during the COVID epidemic.

2.      they went further to recommend that many individuals with prior positive findings on CT should not have further work up during COVID.

3.     they recommended further that certain patients with a diagnosis of stage one lung cancer should have their treatment delayed during the COVID epidemic.

Because the ACCP guidelines for the management of lung cancer are so important in North America, this reference should be included.  I would also ask the authors whether there is any evidence in their survey review to indicate that the ACCP guidelines had any influence on the practice of physicians in their hospital, that may have contributed  to delay in screening, work up and or treatment of patients with lung cancer in this study? 

It is very clear that medical organizations and governments experienced many failures during the COVID era. A rigorous and candid review and critique is necessary to guide future clinicians during the next pandemic.

Author Response

Response to Reviewer 1 Comments

We appreciate the careful review and constructive suggestions. It is our belief that the manuscript is substantially improved after making the suggested edits. 

Below are our responses. The revision has been developed in consultation with all coauthors, and each author has given approval to the final form of this revision. 

Point 1.  This is a very important research study that should be published. Aside from a rare grammatical or typographical error, the manuscript is very well written and requires no revisions.

The only critique that I can offer to the authors, for their possible revision, is the omission of a reference to a guideline article from the American College of Chest Physicians (ACCP) that in 2020, made very specific recommendations.

Response: We agree that with your comment and introduced the citation in the first paragraph in introduction:

An expert panel of 24 members, including pulmonologists, thoracic radiologist and thoracic surgeon have reviewed pre-COVID-19 guideline recommendations for lung screening and lung nodule evaluation[1]. The consensus was that it is appropriate to defer lung cancer screening and modify the evaluation of lung nodules due to risk from potential exposure and the need to relocate resources. Despite the revised recommendations medical organisations experienced many failure during the COVID-era. “

  1. Mazzone, P.J., M.K. Gould, D.A. Arenberg, et al., Management of Lung Nodules and Lung Cancer Screening During the COVID-19 Pandemic: Chest Expert Panel Report. Chest 2020. 158(1): p. 874-8.[PubMed]· https://pubmed.ncbi.nlm.nih.gov/32335067/.

Reviewer 2 Report

Interesting and topical article.

Two suggestions:

Implement the introduction: out of the 2500 articles published and cited, only 3 are listed here. Please add articles as shown below.

The 35% drop in lung cancers does not specify whether there are differences by sex: it could be interesting to understand if the drop concerned only one sex, as in the article cited. The 2020 vs 2019 comparison emphasized a decline in lung cancers in a northern Italian province, but with a strong gender difference. -17.6% in males and + 14.1% in females [Mangone L, Marinelli F, Bisceglia I, Pinto C. The incidence of cancer at the time of Covid in northern Italy. Annals of Research in Oncology, Vol. 2(2), 105-115, 2022]

Introduction

During the pandemic period, an increased risk of mortality was observed in cancer patients, especially in the elderly, males, people with comorbidities, smokers and people with low performance status [Kuderer NM, Choueiri TK, Shah DP, et al. COVID-19 and Cancer Consortium. Clinical impact of COVID-19 on patients with cancer (CCC19): a cohort study. Lancet. 2020 Jun 20; 395 (10241): 1907-1918].

In the UK, mortality in patients with cancer on chemotherapy is associated to higher risk, but not significantly (HR 1.5; 95% CI 0.91-2.45). [Lee LYW, Cazier JB, Starkey T, et al. COVID-19 prevalence and mortality in patients with cancer and the effect of primary tumor subtype and patient demographics: a prospective cohort study. Lancet Oncol. 2020 Oct; 21 (10): 1309-1316.]

The impact of chemotherapy has shown different effects: in patients with lung cancer, there were higher rates of severe or critical COVID-19 events (HR 2.0; 95% CI 1.20-3.30) [Jee J, Foote MB, Lumish M, et al . Chemotherapy and COVID-19 Outcomes in Patients with Cancer. J Clin Oncol. 2020 Oct 20; 38 (30): 3538-3546].

In addition, Italian studies showed an increased risk of being hospitalized and dying from COVID-19 than compared to the general population, in particular for lung, breast and hematological cancers [Rugge M, Zorzi M, Guzzinati S. SARS-CoV-2 infection in the Italian Veneto region: adverse outcomes in patients with cancer. Nat Cancer 2020; 1: 784–788]. However, a previous work showed that for respiratory cancers there was an excess risk of being hospitalized (HR 3.63; 95% CI 1.26-10.44), but not of dying (HR 1.64; 95% CI 0.58-4.64). [Mangone L, Gioia F, Mancuso P, Bisceglia I, Ottone M, Vicentini M, Pinto C, Giorgi Rossi P. Cumulative COVID-19 incidence, mortality and prognosis in cancer survivors: A population-based study in Reggio Emilia, Northern Italy . Int J Cancer. 2021 Apr 16; 149 (4): 820-6. doi: 10.1002 / ijc.33601].

Concerning incidence, an Italian study showed that during the lockdown (March-May 2020) in Italy, there was a 45% reduction in new cancer diagnoses compared with the same months of 2018-2019 [Ferrara G, De Vincentiis L, Ambrosini-Spaltra A, Barbareschi M, Bertolini V, Contato E, Crivelli F, Feyles E, Mariani MP, Morelli L, Orvieto E, Pacella E, Venturino E, Saragoni L. Cancer Diagnostic Delay in Northern and Central Italy During the 2020 Lockdown Due to the Coronavirus Disease 2019 Pandemic. Am J Clin Pathol. 2021 Jan 4; 155 (1): 64-68]. In particular, the decrease concerned skin cancers and melanomas (-57%), colorectal (-47%), prostate (-45%) and lung (-27%) cancer. A subsequent study evaluated the impact that the lockdown (and the suspension of screening tests) had on new cancer diagnoses, highlighting a 35% decrease in new diagnoses, in particular of breast (-35%), prostate (-32%) and lung (-22%) cancer [Mangone L, Giorgi Rossi P, Grilli R, Pinto C. Lockdown Measures Negatively Impacted Cancer Care. Am J Clin Pathol. 2021 Mar 15; 155 (4): 615-616].

In general:

Introduction: to be implemented

Materials: ok

Results: ok

Disucssion: to implement

Tables and figures: ok

It can be published with minor revisions

Author Response

Response to Reviewer 2 Comments

We appreciate the careful review and constructive suggestions. It is our belief that the manuscript is substantially improved after making the suggested edits. 

Below are our responses. The revision has been developed in consultation with all coauthors, and each author has given approval to the final form of this revision. 

Point 1: The 35% drop in lung cancers does not specify whether there are differences by sex: it could be interesting to understand if the drop concerned only one sex, as in the article cited.

Response: The incidence of lung cancer by gender was reported in Table 2.

In the second paragraph of discussion we introduced:We observed a trend of declining of new diagnoses of lung cancer among males and increasing in females. Similarly, Mangole et al. reported strong gender difference in rate of lung cancer diagnosis during the 1st year of COVID when compared to pre-COVID year: 17.6% drop in males and 14.1% increase in females and the reference:

14        Mangone, L., L. Mrinelli, I. Bisceglia, et al., The incidence of cancer at the time of COVID-19 in Northern Italy. Annals of Reserch in Oncology, 2022. 2(2): p. 105-115.[PubMed]· https://www.ncbi.nlm.nih.gov/pubmed/36186539.

Point 2.  Implement the introduction: out of the 2500 articles published and cited, only 3 are listed here. Please add articles as shown below.

Response: All recommended articles are cited and added to the references:

In first paragraph of the discussion we added: “Studies from Italy reported a 20-27% drop in lung cancer diagnoses as an impact of COVID lockdown.”

12         Ferrara, G., L. De Vincentiis, A. Ambrosini-Spaltro, et al., Cancer Diagnostic Delay in Northern and Central Italy During the 2020 Lockdown Due to the Coronavirus Disease 2019 Pandemic. Am J Clin Pathol, 2021. 155(1): p. 64-68.[PubMed]· https://www.ncbi.nlm.nih.gov/pubmed/32995855.

  1. Kumari, S., R.S. Mahla, L. Mangone, et al., Lockdown Measures Negatively Impacted Cancer Care. Am J Clin Pathol, 2021. 155(4): p. 615-616.[PubMed]· https://www.ncbi.nlm.nih.gov/pubmed/33382874.

The remaining recommended citation were introduced in the last paragraph of introduction after:

More than 2500 studies from across the world were reporting hardship with cancer care delivery: many cancer diagnostic tests and treatments were canceled or delayed because of COVID-19, increased risk of contacting COVID and mortality in cancer patients, especially in patients undergoing chemotherapy

  1. Jee, J., M.B. Foote, M. Lumish, et al., Chemotherapy and COVID-19 Outcomes in Patients With Cancer. J Clin Oncol, 2020. 38(30): p. 3538-3546.[PubMed]· https://www.ncbi.nlm.nih.gov/pubmed/32795225.
  2. Kuderer, N.M., T.K. Choueiri, D.P. Shah, et al., Clinical impact of COVID-19 on patients with cancer (CCC19): a cohort study. Lancet, 2020. 395(10241): p. 1907-1918.[PubMed]· https://www.ncbi.nlm.nih.gov/pubmed/32473681.
  3. Lee, L.Y.W., J.B. Cazier, T. Starkey, et al., COVID-19 prevalence and mortality in patients with cancer and the effect of primary tumour subtype and patient demographics: a prospective cohort study. Lancet Oncol, 2020. 21(10): p. 1309-1316.[PubMed]· https://www.ncbi.nlm.nih.gov/pubmed/32853557.

Round 2

Reviewer 1 Report

This is an important contribution that will alert practitioners about the very real and significant morbidity and mortality that will inevitably result from delay in  screening or treatment of lung cancer.